# Modulation of Toll-like receptor 1 intracellular domain structure and activity by $Zn^{2+}$ ions

Vladislav A. Lushpa[1,2], Marina V. Goncharuk[1], Cong Lin[3], Arthur O. Zalevsky [1], Irina A. Talyzina [1,4], Aleksandra P. Luginina[2], Daniil D. Vakhrameev[2], Mikhail B. Shevtsov[2], Sergey A. Goncharuk [1,2], Alexander S. Arseniev[1], Valentin I. Borshchevskiy [2,5,6], Xiaohui Wang[3,7] & Konstantin S. Mineev [1,2 ✉]

Toll-like receptors (TLRs) play an important role in the innate immune response. While a lot is known about the structures of their extracellular parts, many questions are still left unanswered, when the structural basis of TLR activation is analyzed for the TLR intracellular domains. Here we report the structure and dynamics of TLR1 toll-interleukin like (TIR) cytoplasmic domain in crystal and in solution. We found that the TLR1-TIR domain is capable of specific binding of Zn with nanomolar affinity. Interactions with Zn are mediated by cysteine residues 667 and 686 and C667 is essential for the Zn binding. Potential structures of the TLR1-TIR/Zn complex were predicted *in silico*. Using the functional assays for the heterodimeric TLR1/2 receptor, we found that both Zn addition and Zn depletion affect the activity of TLR1, and C667A mutation disrupts the receptor activity. Analysis of C667 position in the TLR1 structure and possible effects of C667A mutation, suggests that zinc-binding ability of TLR1-TIR domain is critical for the receptor activation.

[1] Shemyakin-Ovchinnikov Institute of Bioorganic Chemistry RAS, Moscow, Russia. [2] Moscow Institute of Physics and Technology, Dolgoprudny, Russia. [3] Laboratory of Chemical Biology, Changchun Institute of Applied Chemistry, Chinese Academy of Sciences, Changchun, Jilin, China. [4] Center of Life Sciences, Skolkovo Institute of Science and Technology, Moscow, Russia. [5] Institute of Biological Information Processing (IBI-7: Structural Biochemistry), Forschungszentrum Jülich GmbH, Jülich, Germany. [6] JuStruct: Jülich Center for Structural Biology, Forschungszentrum Jülich GmbH, Jülich, Germany. [7] Department of Applied Chemistry and Engineering, University of Science and Technology of China, Hefei, China. ✉email: mineev@nmr.ru

Toll-like receptors (TLRs) take part in the innate immune response and may serve as targets for the drug design against the inflammatory, neurodegenerative, and autoimmune disorders[1–3]. Human TLR family includes ten members, which can recognize various pathogen-associated molecular patterns[4,5]. Receptors belong to the type I of membrane proteins and contain the large extracellular ligand-binding domain, single-pass transmembrane domain, and globular intracellular Toll-interleukin receptor homology (TIR) domain. The molecular mechanism of TLR activation is believed to be known: ligand binding induces the receptor dimerization, which, in turn, triggers the interaction of TIR domains with intracellular adaptor proteins, namely myeloid differentiation primary response 88 protein (MyD88), TIR domain containing adaptor protein (TIRAP), tumor necrosis factor receptor-associated factors, etc.[6]. In particular, TLR1 is shown to form the heterodimers with TLR2, which are activated upon the binding of lipoteichoic acid or cysteine-containing lipopeptides[7,8]. Either heterodimerization itself or the specific conformation of the dimer, induced by the ligand binding, are believed to cause the interaction between the TIR domains of TLR2 and MyD88, and assembly of a signaling complex, referred to as myddosome[9,10].

TLR extracellular domains are studied rather well—much X-ray data are available, including several structures of dimeric domains in complex with various ligands[11–19]. Nuclear magnetic resonance (NMR) structures of TLR3 and TLR4 transmembrane domains were also obtained[20,21]. Finally, a low-resolution cryoelectron microscopy density (Cryo-EM) map of full-length TLR5 in detergent micelles was reported[22] and computer models of dimeric TLR3 and TLR4 were proposed[23,24]. In the most recent work, structures of full-length TLR3 and TLR7 were solved by Cryo-EM at 3.1 Å resolution in complex with UNC93B1 chaperone; however, the density of the TIR domains was not observed[25]. On the other hand, many questions remain unanswered, if TLR activation is considered from the inside of a cell, whereas four X-ray structures of TIR domains are available (TLR1[26], TLR2[27], TLR6[28], and TLR10[29]). First of all, the reason why the TIR domain would interact with adaptors exclusively in the dimeric state is not clear. For the case of TLR1/TLR2 system, how the association with TLR1 can render the TLR2 binding to MyD88 is still unknown. Second, all the studied TLR TIR domains do not homodimerize in vitro[26]. Except for the TLR10 TIR domain that was shown to be dimeric in crystals, all other three resolved structures of dimeric TLR TIR domains were stabilized by non-native disulfide bonds. To fill these "blank spots," we initiated the investigation of TLR1-TIR domain structure and dynamics in crystal and in solution, focusing on the factors that can influence the interaction between the TIR domains, including the presence of metal ions.

## Results

### Solution structure of TLR1-TIR domain differs from the crystalline conformation and reveals several flexible regions.
The TIR domain of TLR1 (TLR1-TIR) was already studied previously by X-ray diffraction[26]. The protein appeared monomeric in solution, according to the size-exclusion chromatography data. However, the obtained structure was stabilized by a disulfide bond between the Cys residues 667 and 686, which is likely to be non-native. The cytoplasm of eukaryotic cells is highly reducing, due to the presence of glutathione, which should prevent the formation of cysteine bridges[30]. Thus, we decided to investigate the TLR1-TIR in a more native environment and engineered a construct, corresponding to the residues 625–786 of human TLR1[31]. The TLR1-TIR was synthesized in *Escherichia coli* and was kept in the aqueous buffer, containing a potent reducing

agent, tris(2-chloroethyl) phosphate (TCEP), to avoid the disulfide formation during all the purification stages. The resulting protein was then studied by heteronuclear NMR spectroscopy. Despite the relatively large size of the object (19.2 kDa), we obtained the high-quality NMR spectra (Supplementary Figs. 1 and 2), which allowed 95.4% of possible chemical shifts assignment. Using the conventional nuclear Overhauser effect experiments, we gathered 5018 interproton cross-peaks and performed the semi-automated structure calculation[32]. The result is shown in Fig. 1.

The overall fold of the resolved structure is typical for the TIR domains: five α-helices and a five-strand β-sheet (Fig. 1a, b). It is rather well defined by NMR data: root mean square deviations (RMSDs) of backbone coordinates of the structured core is as low as 0.8 Å (Table 1). However, several regions of TLR1-TIR are disordered—BB-loop, CD-loop, several N-, and C-terminal residues. To find out whether the observed disorder does actually take place, or it is a result of the lack of data, we measured the NMR relaxation parameters at two protein concentrations (Supplementary Figs. 3 and 4) and analyzed them using the model-free approach[33]. Relaxation data reveal that all four regions are indeed mobile. Terminal parts and BB-loop are characterized by decreased order parameters, which could be interpreted as motions on the ps–ns timescale (Fig. 1c). BB-loop, helix B, and helices C' and C experience slow motions on the μs–ms scale, which is manifested in the increased exchange contributions to the transverse relaxation $R_{ex}$ (Fig. 1d). Cross-peaks of CD-loop residues are extremely broad, which prevents the relaxation measurement; however, the mere line broadening implies the presence of slow motions. Thus, the disordered regions of NMR structures are in fact mobile in solution.

We have also performed the TLR1-TIR crystallization in the excess of TCEP to prevent the formation of cysteine bridges. Protein crystals appeared several months after settling the crystallization (Supplementary Fig. 5) in two space groups $P6_422$ and $P6_222$. $P6_422$ crystals resemble the previously reported one[26] with a similar diffraction resolution of ~3 Å, whereas $P6_222$ crystals were not published before and provide better diffraction of ~2.5 Å. Protein structures obtained in two space groups are almost similar and both reproduce the previously reported 1FYV including the intramolecular C667–C686 and intermolecular C707–C707' bridges. We used the structure in the $P6_222$ space group for further analysis, because it has a better resolution (see Table 2 for details of X-ray data collection and structure refinement statistics).

The obtained NMR structure can be compared with our X-ray data (Fig. 2). Two structures are mostly similar, the backbones of five β-strands could be superimposed with RMSD of 0.65 Å (Fig. 2a). Two major differences are observed in the position of helix αE and conformation of a BB-loop (Fig. 2b, c). Although the first region was not found to be involved in any known TLR1 activity, the conformation of the BB-loop is essential, as this region is known to participate in TIR–TIR interactions[34,35]. In the X-ray structure, the BB-loop is stabilized by a disulfide bond, which results in the presence of an additional helix turn 669–673. This makes the loop shorter and more compact. In contrast, in solution the disulfide is not formed and the BB-loop is larger and almost completely disordered, which is supported by the relaxation analysis.

### TLR1-TIR binds the Zn²⁺ ions specifically with nanomolar affinity.
Analysis of the solution structure made us consider the possible Zinc-binding propensity of the TLR1-TIR. Two cysteines that engage into a disulfide bond in the X-ray structure of the protein but are reduced in solution are too close to each other and

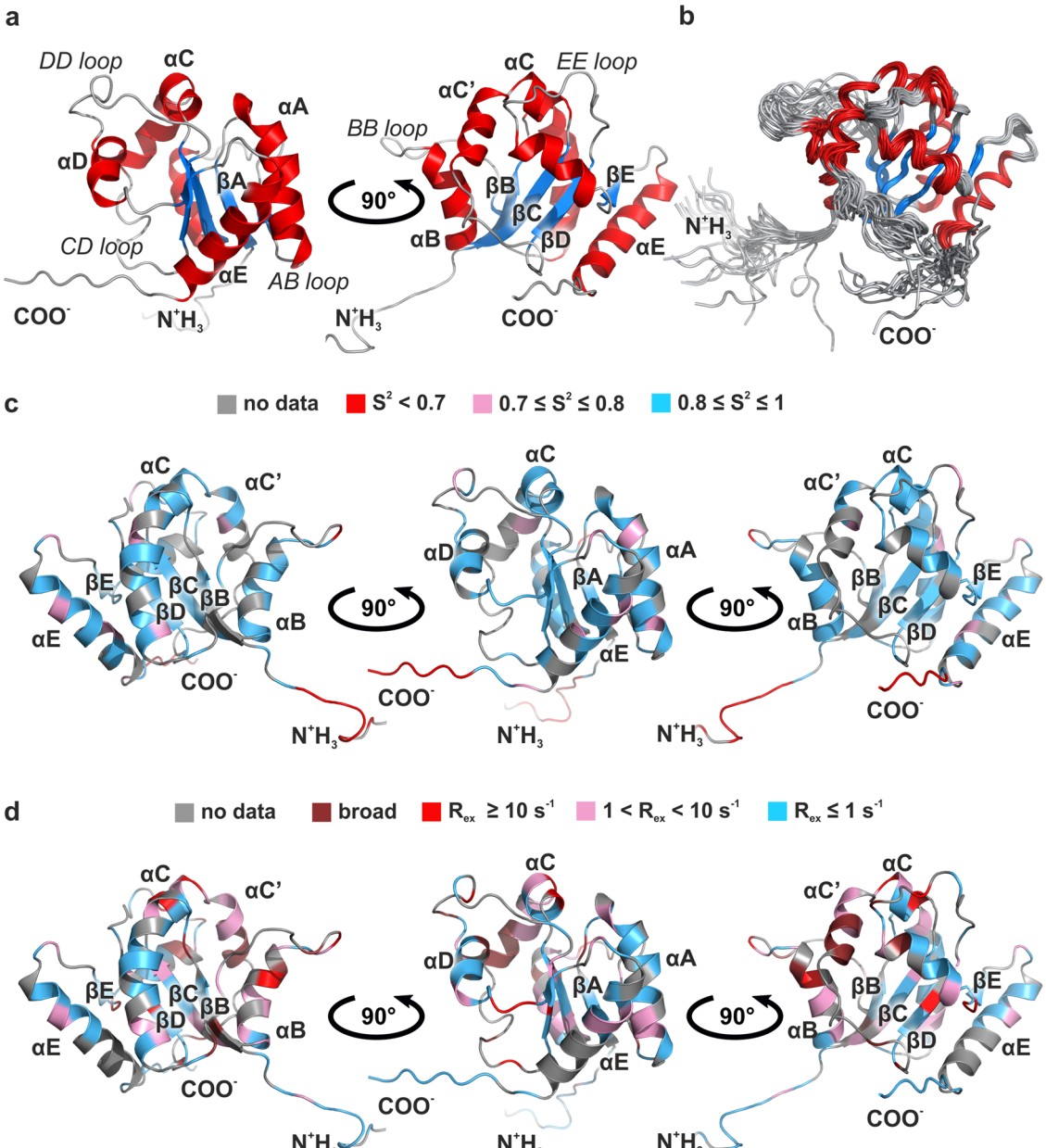

**Fig. 1 Spatial structure and dynamics of TLR1-TIR. a** Protein structure in ribbon representation. **b** Twenty best structures of TLR1-TIR, superimposed over the backbone atoms of the secondary structure elements. α-Helices are colored in red, β-sheets are shown in blue, coil regions are shown in gray. **c** Fast (ps–ns) motions of the protein backbone. Spatial structure of TLR1-TIR is colored with respect to the generalized order parameters of NH bonds. **d** Slow (μs–ms) motions of the protein backbone. Spatial structure of TLR1-TIR is colored with respect to the exchange contribution to the $^{15}$N transverse relaxation. Regions with no relaxation data measured are colored in gray in **c** and **d**. α-Helices, loops, and β-sheets of TLR1-TIR are assigned using the conventional nomenclature accepted for the TIR domains. Regions with signals being observed in the spectrum, but with relaxation parameters that cannot be measured reliably due to the excessive line broadening are marked in brown.

are likely forming a Zinc-binding site, because cysteines are the most frequent residues in zinc coordination spheres[36]. Thus, we titrated a 100 μM TLR1-TIR sample with ZnCl$_2$, choosing the buffer conditions close to the contents of cell cytoplasm. Our data reveal that Zn binds to the TLR1-TIR at 1 : 1 molar ratio, according to the slope of the unbound protein concentration curve (Fig. 3c). Zn binding is slow (characteristic time is >100 ms) and two sets of new signals with equal intensity appear in the NMR spectra of the TIR domain in the presence of metal ions (**Zn1** and **Zn2**, Fig. 3a and Supplementary Fig. 6). This may be interpreted as three different options as follows: (1) the presence of two competing binding sites, (2) the presence of a second

oligomeric state of the protein, or (3) the formation of an asymmetric homodimer upon the Zn binding. To analyze these options, we investigated the hydrodynamic properties of the TLR1-TIR by dynamic light scattering (DLS) and NMR in the presence and in the absence of Zn$^{2+}$ ions (Fig. 3d and Supplementary Fig. 7).

In the absence of Zn$^{2+}$ ions, the TLR1-TIR protein is predominantly monomeric at concentrations up to 400 μM. Hydrodynamic radii are measured by the NMR diffusion and DLS = 2.0–2.2 nm, which corresponds to the globular protein of 16–19 kDa (weight of the TLR1-TIR is 19.2 kDa) (Supplementary Fig. 7). According to DLS, the hydrodynamic radius of the TLR1-

TIR corresponds to the monomeric state up to 1 : 1 metal to protein molar ratio. However, at the threefold excess of zinc, the average hydrodynamic radius raised to 3.78 ± 0.05 nm (~94 kDa, pentamer). This is in agreement with NMR data—at the excess of zinc, cross-peaks vanish in the spectra of TLR1-TIR. Rotational diffusion correlation times in TLR1/Zn 1 : 1 mixture lie in the range 8.6–10.4 ns for two observed Zn-bound states, all values corresponding to the monomeric form of the protein. Therefore, up to 1:1 Zn content, the protein is predominantly monomeric and oligomerizes at the excess of $Zn^{2+}$ ions. According to the

NMR titration, the whole intensity of the initial cross-peak of TLR1-TIR is equally distributed between the two newly formed Zn-bound states; no third state with two zinc ions bound is detected. Thus, the two observed states of TLR1-TIR/Zn complex correspond to the alternative modes or competing sites for the Zn binding.

It is noteworthy that Zn binding by TLR1-TIR is reversible: the initial state of the protein is restored by the addition of a potent chelator, such as ethylenediaminetetraacetate (EDTA). Thus, to measure the Zn-binding propensity, we applied the chelator

### Table 1 NMR and refinement statistics for protein structures.

|  | TLR1-TIR |
| --- | --- |
| *NMR distance and dihedral constraints* | |
| Distance constraints | |
| Total NOE | 1995 |
| Intra-residue | 669 |
| Inter-residue | 1326 |
| Sequential ($\|i − j\| = 1$) | 570 |
| Medium-range ($\|i − j\| < 4$) | 354 |
| Long-range ($\|i − j\| > 5$) | 402 |
| Hydrogen bonds | 102 |
| Total dihedral angle restraints | 291 |
| $\phi$ | 136 |
| $\psi$ | 136 |
| $\chi_1$ | 19 |
| *Structure statistics* | |
| Violations (mean and SD) | |
| Distance constraints (Å) | 0.0167 ± 0.0016 |
| Dihedral angle constraints (°) | 1.37 ± 0.068 |
| Max. dihedral angle violation (°) | 13.96 |
| Max. distance constraint violation (Å) | 0.48 |
| Deviations from idealized geometry | |
| Bond lengths (Å) | 0 |
| Bond angles (°) | 0 |
| Impropers (°) | 0 |
| Average pairwise r.m.s. deviation[a] (Å) | |
| Heavy | 1.31 ± 0.16 |
| Backbone | 0.80 ± 0.18 |

[a]Pairwise r.m.s. deviation was calculated among 20 refined structures.

### Table 2 Data collection and refinement statistics (molecular replacement).

|  | TLR1-TIR w/o $Zn^{2+}$ | TLR1-TIR with $Zn^{2+}$ 1 : 1 |
| --- | --- | --- |
| *Data collection* | | |
| Space group | P 62 2 2 | P 62 2 2 |
| Cell dimensions | | |
| $a, b, c$ (Å) | 101.26, 101.26, 68.85 | 101.52, 101.52, 68.86 |
| $\alpha, \beta, \gamma$ (°) | 90, 90, 120 | 90, 90, 120 |
| Resolution (Å) | 50.63–2.47 (2.558–2.47)[a] | 43.96–1.9 (1.968–1.9) |
| $R_{sym}$ or $R_{merge}$ | 0.1806 (1.867) | 0.103 (5.126) |
| $I/\sigma I$ | 21.19 (2.24) | 22.88 (0.69) |
| Completeness (%) | 99.92 (100.00) | 99.73 (99.04) |
| Redundancy | 37.5 (39.2) | 38.2 (34.8) |
| *Refinement* | | |
| Resolution (Å) | 50.63–2.47 | 43.96–1.9 |
| No. reflections | 7875 | 16,969 |
| $R_{work}/R_{free}$ | 0.2291/0.2791 | 0.2151/0.2455 |
| No. atoms | 1313 | 1330 |
| Protein | 1302 | 1291 |
| Ligand/ion | 0 | 0 |
| Water | 11 | 39 |
| *B-factors* | 55.61 | 57.34 |
| Protein | 55.67 | 57.40 |
| Ligand/ion | – | – |
| Water | 48.29 | 55.32 |
| R.m.s. deviations | | |
| Bond lengths (Å) | 0.001 | 0.003 |
| Bond angles (°) | 0.40 | 0.50 |

[a]Values in parentheses are for highest-resolution shell.

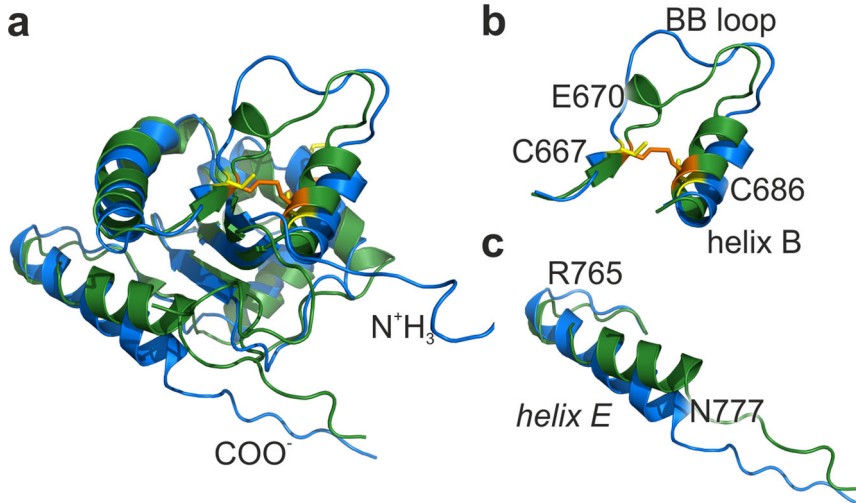

**Fig. 2 Superposition of NMR and X-ray structures. a** Overlay of TLR1-TIR structures obtained by NMR spectroscopy (blue and yellow) and X-ray crystallography (green and orange). **b, c** Overlay of TLR1-TIR regions with the most significant structural changes caused by the closure of a disulfide bond (yellow and orange colors, respectively).

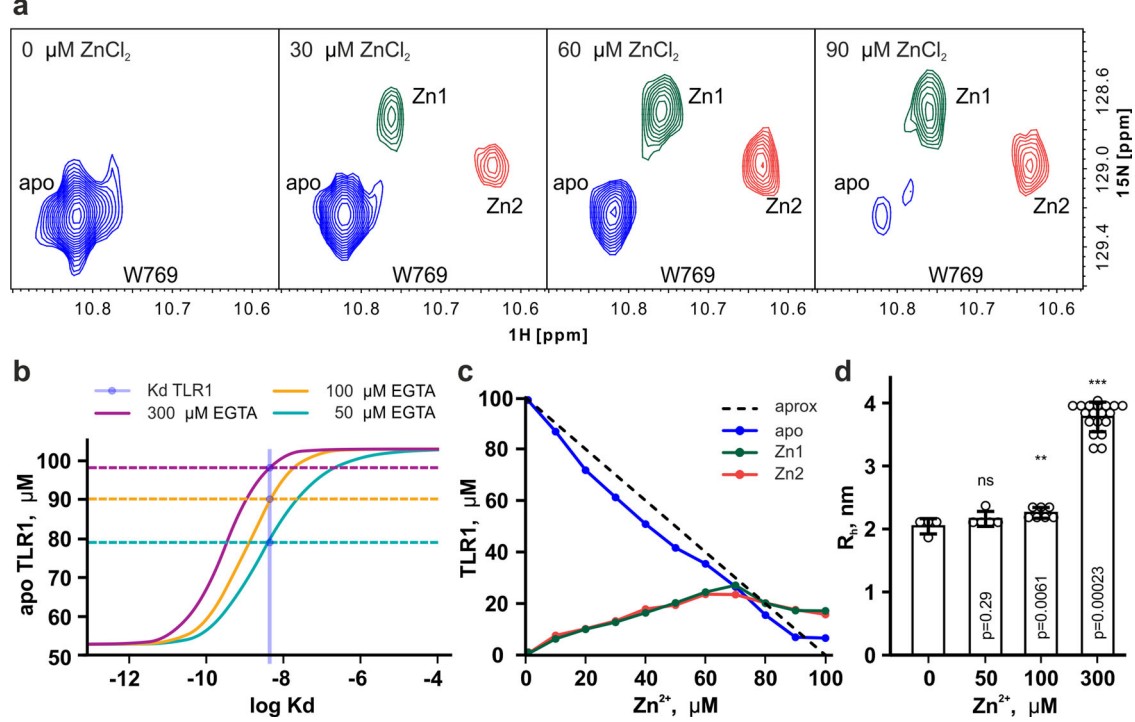

**Fig. 3 Zn binding by TLR1-TIR. a** Fragments of $^1$H,$^{15}$N-HSQC spectra of TLR1-TIR. Spectra were recorded after the addition of 0, 30, 60, and 90 μM ZnCl$_2$ to the 100 μM sample of TLR1-TIR at 35 °C and pH 7.4. Signals of W769 indole NH (in blue) and two sets of signals in the presence of metal ions (in green and yellow, assigned as Zn1 and Zn2) are shown. **b** Results of EGTA competition assays. Solid lines represent the theoretical dependencies predicted for the Zn-unbound state of the TLR1-TIR as a function of log(Kd) at the corresponding concentration of EGTA. Dashed lines represent the measured concentration of Zn-free TLR1-TIR. Blue region denotes the intersections of dashed and solid lines, and corresponds to the measured Kd range. The sample contained 100 μM TLR1-TIR and 50 μM ZnCl$_2$. **c** Concentrations of Zn-unbound TLR1-TIR (apo) and of two Zn-bound states (Zn1, Zn2) as a function of ZnCl$_2$ concentration in solution. The dashed line represents the $y = 100 - x$ function, expected for the 1 : 1 binding. **d** Hydrodynamic radii of the TLR1-TIR measured by DLS in 100 μM solution at various concentrations of ZnCl$_2$. *p*-Values are provided, according to the Mann–Whitney test. Error bars represent SD.

competition assay. We observed that TLR1-TIR cannot compete with EDTA (pKd = 13.6) but competes with egtazic acid (EGTA) (pKd = 9.2)[37], which allowed determining the TLR1-TIR/Zn stability constant equal to 4.5 ± 0.5 nM (Fig. 3b). To examine the specificity of Zn binding, we performed similar experiments with other divalent metal cations that are widely spread in the cell cytoplasm: Ca, Mg, Fe, Mn, Co, Ni, and Cu. It appeared that Ca, Mg, Fe, Mn, and Ni did not change the NMR spectra of TLR1-TIR. Co and Cu addition, in contrast, resulted in drastic changes (Supplementary Fig. 8). On the other hand, NMR spectra of TLR1-TIR in the presence of Co and Cu were completely identical, which is impossible if the metal is bound, taking into account the difference in paramagnetic properties of these two metals (Supplementary Fig. 9). Further analysis revealed that the spectrum in the presence of Co/Cu corresponds to the TLR1-TIR state with the formed C667–C686 disulfide bridge (Supplementary Fig. 10). Thus, although Mg, Fe, Mn, and Ni do not bind to TLR1-TIR, Co and Cu addition catalyzes the disulfide bond formation, which is now formed even in the presence of the reducing agent. Further, only the Zn$^{2+}$ ions are bound by TLR1-TIR specifically, reversibly, and with nanomolar affinity.

**C667 is a critical residue of TLR1-TIR Zn-binding site.** At the excess of Zn, the quality of NMR spectra drops dramatically, revealing further oligomerization. Moreover, the protein becomes much less stable in the presence of Zn and tends to precipitate. These factors do not allow obtaining the sample of a Zn-bound TLR1-TIR at a high concentration, necessary for the NMR chemical shift assignment and structure determination. We

undertook several attempts to crystallize the TLR1-TIR/Zn complex. Similar to TLR1-TIR without Zn, crystals appeared in several months in P6$_4$22 and P6$_2$22 space groups, where the last one gave better diffraction. The best crystal diffracted to 1.9 Å and was used for the structure solution (Table 2). All the crystals that were obtained provided the electron density with the Zn$^{2+}$ ions absent and all cysteine residues in the oxidized state, similar to the structure of TLR1-TIR without Zn. We assume that the disulfide crosslinked conformation of TLR1-TIR is the only state of the protein capable of crystallization.

As all the direct approaches failed to localize the binding site, we applied the point mutagenesis and synthesized the C707A, C686A, and C667A variants of TLR1-TIR. The mutants were titrated by Zn and NMR spectra were recorded. All the mutants did not change the overall fold of TLR1-TIR: the position of characteristic NMR cross-peaks of most amide and methyl groups was retained (Supplementary Figs. 11 and 12). According to the obtained data, C707A mutation does not affect the Zn binding, whereas C667A completely abolishes the interaction. In the case of C686A substitution, the binding is retained; however, only one Zn-bound state of the protein is observed (**Zn2**) instead of two for the wild-type (WT) TLR1-TIR (Fig. 4a). Therefore, C667 is the key residue responsible for the Zn coordination, which takes part in both modes of Zn binding. C686 also participates in the interaction, providing one mode of Zn binding.

**Computer modeling reveals two possible Zn-binding modes in TLR1-TIR BB-loop.** To find the other possible Zn-coordinating residues and to understand the effects of Zn binding on the

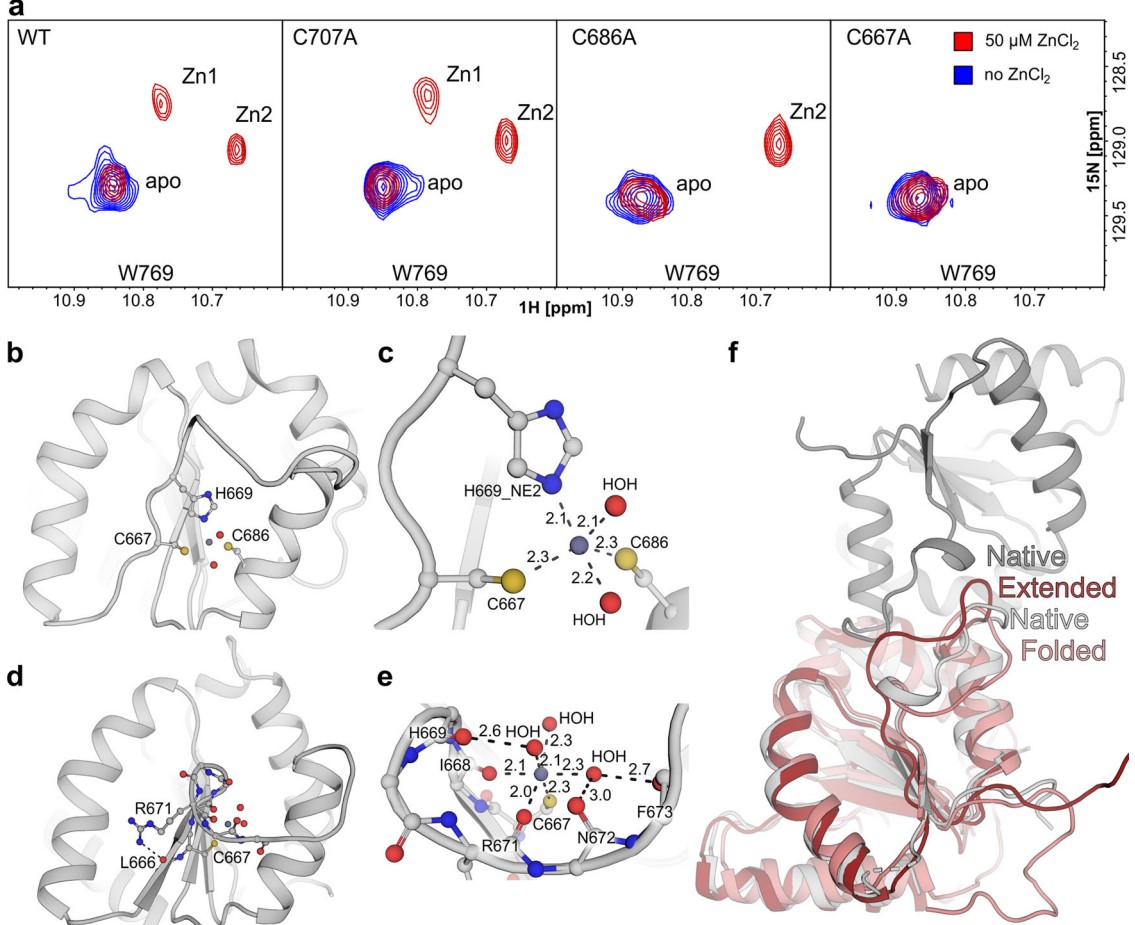

**Fig. 4 Localization of TLR1-TIR Zn-binding site and the model of the Zn-bound state. a** An overlay of $^1$H,$^{15}$N-HSQC spectra (regions with the signal of W769 indole NH) of TLR1-TIR and its mutants C707A, C686A, and C667A. Spectra were recorded before (in blue) and after (in red) the addition of 50 μM ZnCl$_2$ to the 100 μM sample of TLR1-TIR at 30 °C and pH 7.4. Signals corresponding to the Zn-free and two Zn-bound states of TLR1-TIR are assigned as apo, Zn1, and Zn2, respectively. **b** Snapshot from the simulation of the first coordination mode formed by C667-H669-C686. **c** Closeup view of the coordination sphere for the C667-H669-C686 coordination mode. **d** Snapshot from the simulation of second coordination mode formed by C667 and I668_O. Additional coordinators might be represented by the R671_O and water molecules also coordinated by the backbone oxygen of the BB-loop residues. **e** Closeup view of the Zn2 coordination sphere. **f** Comparison of the "extended" and "folded" BB loop conformations with the "native" BB-loop conformation in the TLR10 homodimer.

structure of TLR1-TIR, we turned to computer modeling and bioinformatics. As the BB-loop of TLR1-TIR is flexible and its conformations may be poorly sampled in the NMR ensemble, we first generated the set of BB-loop states using ROSETTA and searched for the potential Zn ligands that may be proximal to the C667 and/or C686 thiol groups (Supplementary Fig. 13). Such an analysis provided only the H669 side chain as a possible coordinator of Zn$^{2+}$ ions. As a next step, we ran several molecular dynamics (MD) simulations, utilizing the force field, optimized for the studies of protein–Zn interactions[38]. At the start point, Zn was placed close to the C667 sulfur atom and the final configuration of the complex was analyzed. Such a procedure provided two distinct modes of Zn binding, which surprisingly correlates well with the NMR data. The first mode is through a classical CCH-binding motif[39] formed by C667, C686, and H669 (Fig. 4b, c). This mode most likely corresponds to the **Zn1** state, which is dependent on the C686 side chain. The second mode is peculiar: the Zn$^{2+}$ ion is coordinated by the thiol group of C667, by the backbone carbonyls of L668 and/or R671, and by the oxygen of a water molecule, which is additionally stabilized by the H-bonds with the backbone carbonyls of other BB loop residues (Fig. 4d, e). This mode could be considered as a candidate for the **Zn2**

state. Two modes correspond to the distinct conformations of the BB-loop (Fig. 4c, e). First mode provides the "extended" conformation of the loop, whereas in the second mode, the loop is "folded" and its structure is stabilized by the additional H-bond between the side chain of R671 and the backbone of I666. Apart from the structure, Zn binding affects the dynamics of the BB loop. Although the atom coordinates root mean square fluctuations (RMSFs) are left unchanged for the major part of TLR1-TIR upon the Zn binding, RMSF drops for several residues of the BB loop (Supplementary Figs. 14 and 15). Moreover, Zn binding narrows the overall conformational landscape (number of possible states) of the BB loop (Supplementary Figs. 14 and 15).

**C667 is the key residue for the TLR1 functionality**. To further investigate the role of Zinc binding in the TLR1 activity, we performed several functional tests in HEK Blue 293 cells, expressing TLR1 and TLR2 receptors. The activity of nuclear factor-κB (NF-κB) was monitored by Phospha-Light secreted embryonic alkaline phosphatase (SEAP) reporter gene assay system after the stimulation of the TLR1/2 receptor with its specific ligand, Pam$_3$CSK$_4$. First of all, we investigated the effect of Zn$^{2+}$ ions on the receptor, by either adding the Zn$^{2+}$ to the

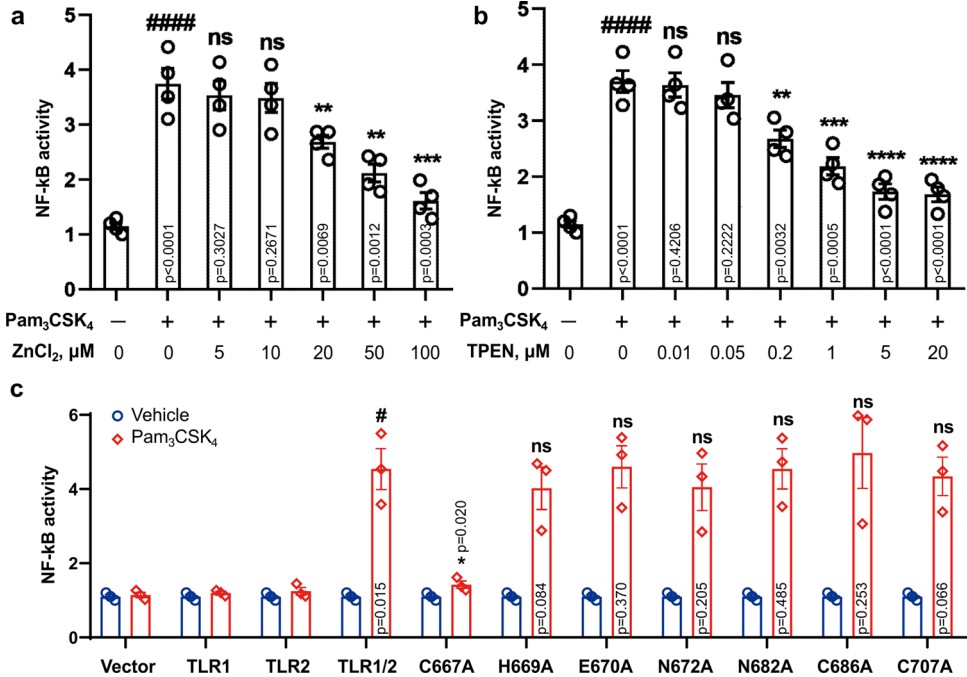

**Fig. 5 Role of Zn binding in TLR1 activity. a, b** NF-κB activity measured upon stimulation of HEK Blue 293 cells transfected with TLR1 and TLR2 genes with Pam₃CSK₄, with the addition of either 0–100 μM ZnCl₂ or 0–20 μM of TPEN to the culture ($n = 4$ independent experiments). **c** NF-κB activity measured for TLR1 mutants upon stimulation with Pam₃CSK₄ ($n = 3$ independent experiments). Statistical significance is indicated as follows: $*p < 0.05$, $**p < 0.01$, $***p < 0.001$, and $****p < 0.0001$ with respect to the positive control, $\#p < 0.05$ and $\#\#\#\#p < 0.0001$ with respect to the negative control experiments. ns denotes that changes with respect to the positive control are not significant. Error bars represent the SEM.

cells from the outside or by removing the free Zn²⁺ ions inside the cell, adding the membrane-permeable zinc-chelating agent N,N,N′,N′-tetrakis(2-pyridinylmethyl)-1,2-ethanediamine (TPEN) (Fig. 5a, b and Supplementary Data 1). Our results show that supplying Zn to the cells inhibits the TLR1/2 activity in a concentration-dependent manner, as well as the Zn depletion of the cell cytoplasm caused by adding the indicated quantities of TPEN. Thus, the presence of Zn²⁺ ions should be considered as an important factor of TLR1 activity; however, the significance of direct interaction between the Zn²⁺ ions and TLR1-TIR is not yet confirmed by this experiment.

Therefore, we next investigated the possible role of several TLR1-TIR residues and assayed the activity of single-point alanine mutants of TLR1 (Fig. 5c, Supplementary Fig. 16, and Supplementary Data 1). Namely, C667A, C686A, C707A, H669A, E670A, N672A, and N682A were selected. Four last residues belong to the BB-loop and were chosen, as their side chains can possibly coordinate the metal ions. Out of seven mutants, only C667A was characterized by an almost complete loss of ligand-induced activity. The other six mutations did not provide any statistically significant changes of NF-κB activation. Thus, we conclude that C667, the residue that is responsible for the Zn binding by TLR1-TIR, is also the key residue for the TLR1/2 receptor functioning.

## Discussion

Summarizing the work, we can list several major findings. First, we determined the solution structure of the TLR1-TIR, which revealed several major differences with respect to the X-ray conformation. These occur mainly in the BB-loop, which is known to be important for the protein–protein interactions of TIR domains[34,35]. All the deviations arise due to the non-native disulfide linkages that take place upon crystallization. According to the original X-ray work, crystallization was run in the presence of a reducing agent, dithiothreitol, which should prevent cysteine

oxidation[26]. In our hands, the protein was as well crystallized with the disulfide bonds, being formed in the presence of TCEP agent, whereas no disulfide bonds were observed in several weeks after the sample preparation for the soluble fraction of TLR1-TIR in solution under the similar conditions. It is noteworthy that crystal growth was essentially slow and took several months, and all the reducing agents have a limited lifetime. Thus, it is most likely that the protein crystallization started at a time point when no active TCEP was present in all the buffers, allowing the formation of cysteine bridges inevitably observed in crystallographic structures and necessary for the crystal packing.

As we show here, cobalt and copper may catalyze the C667–C686 disulfide bond formation, suggesting that the disulfide-cross-linked state can still be physiologically relevant, provided that this process takes place under physiological concentrations of any of two metals. The mechanism of this oxidation is not clear. Cu(II) is known to catalyze the disulfide oxidation, via the redox reaction, accompanied by the copper reduction to Cu(I). On the other hand, a similar reaction for Co(II) has never been reported. Therefore, it is most likely that Co and Cu bind to the TLR1-TIR, forcing it to adopt the conformation that favors the disulfide formation. Analysis of NMR data obtained for TLR1/Co/Cu mixtures reveals that the oxidation process is slow and takes 1–2 h at 100 μM of metal, and that the Co- or Cu-bound state of TLR1 is low-abundant; only the apo-state and the disulfide-crosslinked state of TLR1-TIR are observed in solution. The latter implies that the binding constants of Co and Cu are above 100 μM; otherwise, we would observe the metal-bound state and disulfide-crosslinked state. The concentration in the current work is at least several orders of magnitude higher than the native levels of these metals in cell cytoplasm[40]. Therefore, under the physiological concentrations of metals, the Co/Cu-bound states of TLR1 would be low-abundant and the Co/Cu-induced disulfide oxidation would run extremely slowly, with characteristic times exceeding months. The absence

of the disulfide bonds in the native protein is supported by our single-point mutagenesis analysis. Out of three cysteine residues, only C667 substitution had an effect on the TLR1 activity, which is impossible if C667 is engaged in the disulfide bridge with C686. According to the literature, unnatural disulfide crosslinking in crystals is not a unique event and takes place for several other TIR domains. For instance, disulfide bonds were found in the TIR domain of the TIRAP adapter protein[41]. Later, the solution study revealed that these disulfides are not formed in vitro under the reducing conditions and in vivo inside the cells[42]. In the absence of C667–C686 linkage, the BB-loop of TLR1-TIR domain lost the turn of an α-helix and became flexible and unstructured, according to the NMR structure and dynamics analysis. This conformation of BB-loop should be thus considered as native and used in further modeling/docking experiments.

The most important finding of the work is the ability of TLR1-TIR to bind $Zn^{2+}$ ions. As we show here, TLR1-TIR binds Zn reversibly and specifically with the affinity of 4.5 nM, and the binding site is formed by the residues C667 and C686, wherein the presence of C667 is essential. Despite the fact that concentration of unbuffered Zn inside the cell is thought to lie in the range 10–100 pM, the overall concentration of Zn is above 1 mM[40]; therefore, various proteins compete for the $Zn^{2+}$ ions with each other and low-molecular-weight compounds, such as citrate, cysteine, and glutathione[37]. Here, 4.5 nM is a value that corresponds to the Zn stability constants of several well-known zinc-finger proteins[43], suggesting that TLR1-TIR Zn interaction is physiologically relevant, and in the cytoplasm; at least a part of TLR1 is in a Zn-bound state. It is noteworthy that Zn is a known secondary messenger, which is involved in the activity of many cytoplasmic proteins[44]. In particular, the TLR4 receptor was shown to be activated by free $Zn^{2+}$ ions[45,46]. However, Zn was proposed to be necessary for several events of the downstream signaling cascade and the direct interaction of Zn with TLR4 was not reported[47]. Moreover, stimulation by ligands of several TLRs, including the TLR1/2, resulted in an increase of free Zn concentration in the cytoplasm[48]. According to the data reported here, both Zn excess and Zn depletion of the cell cytoplasm alters the activity of TLR1/2 heterodimer in response to its specific ligand. Thus, Zn is definitely somehow involved in TLR1/2 signaling, the affinity of Zn binding by the TIR domain is high enough to consider the direct interaction of TLR1 with zinc as a possible explanation of Zn-related effects, observed in the in vitro studies.

To further investigate the role of TLR1/Zn interaction, we examined the effect of potential Zn-binding residues on the receptor biological activity. We found out that C667 is one of the key residues of TLR1. C667A mutation disrupts the TLR1/2 ligand-induced activity, whereas all six other mutated residues, including four adjacent amino acids of the BB-loop and two other cysteines of TLR1, did not affect the receptor functionality. It is necessary to point out that functionally important regions of TLR1 are rather poorly studied by mutagenesis. Although a variety of TLR2 regions are already scanned[49,50], only two mutations in TLR1 were shown to alter the receptor behavior. Namely, N672D mutation enables TLR1 to interact with MyD88 adapter protein[51]; however, as we show here, N672A substitution does not affect the TLR1 function. Besides, G676L substitution inhibits the TLR1/2 activity[50]. Thus, C667 is the second residue, found important for the TLR1 functioning, which is definitely a significant result. The essential role of C667 correlates perfectly with the Zn-binding activity of TLR1-TIR—C667 is the sole cysteine of the domain that switches off the Zn uptake. We need to note that C667A mutation is almost synonymous in terms of the side chain hydrophobicity[52] and C667 side chain does not take part in any intramolecular hydrogen bonding or other stabilizing interactions, except for the

aromatic-thiol π-type contact with the F637 ring. Changes caused by the C667A mutation in the NMR spectra of TLR1-TIR are less pronounced than the ones caused by C686A and are located more compactly on the spatial structure of the protein, implying that the structure of the mutant domain is not changed substantially compared to the WT protein (Supplementary Fig. 17). The effect of C667A mutation is also unlikely to be caused by some redox reaction, important for the receptor activation. The environment of cell cytoplasm is highly reducing and redox reactions were never reported to participate in TLR activation. Finally, only 15% of the C667 area is solvent-accessible in our NMR structure, indicating that this residue is not available for the intermolecular contact. In other words, C667A mutation does not affect the structure of the TLR1-TIR domain and is unlikely to directly partake in TLR1-TIR interactions with TLR2 or adapter proteins. Thus, we conclude that the essential role of C667 is related to the Zn-binding ability of the TLR1-TIR domain and the Zn-bound state of the TLR1-TIR should correspond to the functionally active receptor conformation.

The mechanism of how Zn binding by C667 is involved in TLR1 activation has yet to be elucidated. However, we could use the results of computer modeling to speculate about the structure of TLR1-TIR in the Zn-bound active state. As we mentioned above, we found two modes of Zn binding in MD simulations and NMR spectra. The first mode employs the classical CCH motif and most likely corresponds to the **Zn1** state in our NMR spectra, which is dependent on both the C667 and C686 side chains. According to the functional assay, C686 and H669 substitutions do not affect the TLR1 signaling; therefore, the **Zn1** mode is not important for the activation. The second mode is rather peculiar, as only the C667 side chain is involved in Zn binding. The other coordination bonds are provided by the backbone carbonyls and a water molecule, stabilized by hydrogen bonds[53]. This mode agrees well with the **Zn2** state in NMR spectra (it is independent of the C686 side chain) and with the results of the functional assay—all the mutations tested, except for the C667A, should not affect the Zn binding via this mechanism. Thus, we could assume that the **Zn2** state and predicted Zn-binding mode could correspond to the signaling-active state of the TLR1 receptor. It is also noteworthy that, according to MD simulations, the described binding mode stabilizes the "folded" conformation of the BB-loop, which is close to the state of the loop in the X-ray structure of TLR10 TIR homodimer (Fig. 4f)[29]. Summarizing the data, we put forward a hypothesis in which $Zn^{2+}$ ions can bind to the TLR1-TIR domain BB-loop in **Zn2** mode and stabilize the conformation of the domain, which is capable of intermolecular interactions with TLR2 TIR domains or TIR domains of intracellular adaptor proteins (MyD88/TIRAP, etc.). Although the hypothesis is speculative and relies mainly on the in silico prediction, it is in a good agreement with all the data reported.

To conclude, here we provide the solution NMR structure and internal dynamic parameters of the TLR1-TIR domain with the native conformation of a BB-loop. We show that the TIR domain is capable of specific and reversible binding of zinc ions with nanomolar affinity, with two modes of the binding being observed. Interactions with Zn are mediated by C667 and C686 residues; C686 is responsible for one of the binding modes, whereas C667 is essential for any kind of binding. Potential structures of the TLR1-TIR/Zn complex were predicted in silico. By monitoring the activity of NF-κB upon the TLR1/2 ligand stimulation, we found that both Zn addition and Zn depletion affects the activity of TLR1, and C667A mutation, unlike the substitution of several other residues, disrupts the receptor activity. Analysis of C667 position in the TLR1 structure and possible effects of C667A mutation suggests that zinc-binding ability of TLR1-TIR domain is critical for the receptor activation.

## Methods

**Protein expression/purification**. A gene, expressing the TLR1-TIR (UNIPROT ID: Q15399, residues 625–786), was synthesized by TwistBioscience (USA) and cloned into the pGEMEX-1 vector with the gene encoding the N-terminal His-tag and thrombin cleavage site (MHHHHHHGSGSGLVPRGS). C667A, C686A, and C707A mutations were introduced by PCR using the chemically synthesized oligonucleotides (Evrogen, Russia) and confirmed by DNA sequencing.

Protein was synthesized in *E. coli* BL21(DE3)pLysS strain; details of the production and cell lysis are published in our previous work[31]. The protein was purified taking into account the previously published protocol[54]. Briefly, the cell pellet was resuspended in buffer (pH 7.0, 30 mM 3-(N-morpholino) propanesulfonic acid (MOPS), 250 mM NaCl, 200 mkM phenylmethylsulfonyl fluoride, 0.5% Triton X-100, 2 mM TCEP, 5% glycerol), lysed by ultrasonication on ice until complete cell lysis took place and centrifuged at 15,000 × g, 4 °C for 1 h. After filtration through the membrane with 0.22 μm pore size, the TLR1-TIR was purified by immobilized metal affinity chromatography (IMAC, Ni-sepharose HP resin), accompanied with overnight on-column digestion with 30 units of thrombin (Tekhnologiya Standart, Russia) per 1 mg of hybrid protein at 4 °C. After purification, the buffer was exchanged by Illustra NAP-25 column (Cytiva) to the appropriate NMR buffer (see "Sample preparation"). To obtain a 100–600 μM purified TLR1-TIR for NMR applications, the protein sample was concentrated using the Amicon Ultra 10 K centrifugal filter unit with thorough mixing after each 4 min of spinning at 5000 × g at 4 °C. For the DLS experiments, an additional protein purification via size exclusion chromatography was implemented. The TLR1-TIR was concentrated in the presence of 20% glycerol to the 0.5 ml of 200 μM protein sample using the Amicon Ultra 10 K centrifugal filter unit and applied on the Superdex-75 10/300 GL column (GE) equilibrated with cytoplasm-like buffer (see below). Glycerol was used only during the cell lysis and IMAC, unless otherwise stated. All the buffers contained 0.5–2 mM TCEP and all purification procedures were run on ice or at 4 °C.

**Sample preparation**. NMR structural analysis of 600 μM TLR1-TIR was performed in the buffer, containing 30 mM MOPS pH 6.3, 5 mM TCEP, 25 mM NaCl. Then, 100% D$_2$O was added to the sample to a H$_2$O/D$_2$O ratio of 95 : 5.

To study the TLR1-TIR interaction with metal ions, 100 μM TLR1-TIR was transferred into the cytoplasm-like buffer (30 mM MOPS pH 7.4, 64.4 mM KCl, 5.3 mM NaCl, 0.5 mM MgCl$_2$, 0.5 mM TCEP, 0.001% NaN$_3$) designed to properly mimic the conditions of cellular cytoplasm. D$_2$O was added to the sample to a H$_2$O/D$_2$O ratio of 95 : 5.

**NMR spectroscopy**. NMR spectra were recorded on Bruker Avance III 800 and 600 MHz spectrometers, both equipped with the triple-resonance cryogenic probe (Bruker Biospin, Germany) at pH 6.3 (unless otherwise specified) and temperatures of 25 °C, 35 °C, and 45 °C. Backbone and side chain of the protein were assigned manually using the following NMR spectra: three-dimensional (3D) HNCA, HNCO, HNcoCA, HNcaCO, HNcocaCB, 3D 15N-nuclear Overhauser effect spectroscopy (NOESY)-heteronuclear single quantum coherence (HSQC), HCCH-total correlation spectroscopy, HCCH-correlation spectroscopy (COSY), two-dimensional (2D) CaCo. The triple resonance experiments were recorded using the BEST-transverse relaxation optimized spectroscopy (TROSY) pulse sequences[55]. Aromatic side chains were assigned using the (H)CCH-COSY and (Hb)Cb(CgCC)H experiments. Most spectra were recorded with non-uniform sampling and processed using the compressed sensing approach in the qMDD software (IST algorithm with virtual echo)[56]. $^3J_{C\gamma C}$ and $^3J_{C\gamma N}$ couplings were measured using the spin-echo difference constant time-HSQC spectra, as described[57,58].

NMR spectra were analyzed in the CARA 1.9.1 software. Structure was solved in the CYANA 3.97 program[32] based on NMR data, using the automated procedure of NOESY cross-peak assignment, followed by the manual analysis of the remaining peaks. The φ/ψ torsion angle restraints were obtained by the chemical shift analysis with TALOS-N web-server[59]. The χ$_1$ angle restraints were derived by the manual analysis of vicinal $J$-couplings. Hydrogen bond restraints were added on the final stage of the structure calculation based on the hydrogen/deuterium exchange rate of amide groups. Protein sample was dried and redissolved in 100% D$_2$O solution and amide groups with cross-peaks that remained in the NMR spectra 30 min after dissolving were considered as potential hydrogen bond donors. Spatial structures were analyzed using the MOLMOL software[60] and the PyMOL (Schrodinger, LLC). The NMR structure of TLR1-TIR has 84.5% amino acids in Ramachandran-favored region and 15.5% amino acids in Ramachandran-allowed region.

$^{15}$N relaxation parameters (R$_1$, R$_2$, steady-state NOE, $\eta_{xy}$) were measured at 800 MHz for 400 μM $^{15}$N-labeled TLR1-TIR sample (pH 6.3, 35 °C) using the pseudo-3D experiments acquired in the interleaved mode[61]. The $^{15}$N/$^1$H cross-correlated relaxation rates ($\eta_{xy}$) were measured using 2D $^1$H-$^{15}$N-TROSY-HSQC spectra with the modulation of signal intensity[62]. Relaxation data were analyzed using the Tensor 2.0 program using the Lipari-Szabo model-free approach[33].

Lateral diffusion of TLR1-TIR was measured using the PGSTE-watergate pulse sequence with the suppression of convection[63]. Methyl group region of $^1$H NMR spectrum (1–0.5 p.p.m.) was used for the analysis.

**DLS experiments**. DLS experiments were performed at 10 °C using the Wyatt DynaPro Titan spectrometer. Forty microliters of 100 μM TLR1-TIR was placed in the quartz cuvette, ZnCl$_2$ was added to the protein to obtain 2 : 1, 1 : 1, and 1 : 3 protein to metal molar ratio. At each point, we performed at least four successive measurements, consisting of one hundred 5 s acquisitions. Data were analyzed with DYNAMICS 6.7.6 Software. Derived hydrodynamic radii were averaged among measurements and the significance of the observed differences was estimated using the Mann–Whitney test.

**Metal-binding assay**. Here, 1 M, 100 mM, 10 mM solutions of MnCl$_2$, CaCl$_2$, ZnCl$_2$, NiCl$_2$, CoCl$_2$, CuSO$_4$, and FeCl$_2$ were used as the source of divalent cations (all from Merck, USA). To study the metal binding, a 100 μM sample of $^{15}$N-labeled TLR1-TIR in a cytoplasm-like buffer was titrated with the metal solution and $^1$H,$^{15}$N-HSQC spectra were acquired. Experiments were run at 30 °C to avoid protein precipitation. In the case of Zn, the titration was accomplished with 10 μM steps (the protein : metal ratio varied from 10 : 1 to 1 : 2). To test the other metals, the titration was accomplished with a step of 50 μM (the protein : metal ratio was varied from 2 : 1 to 1 : 2). To quantify the results, intensities of cross-peaks corresponding to the W769 indole NH-group were monitored.

To estimate the dissociation constant of the protein/Zn complex, the sample, containing 100 μM of TLR1-TIR and 50 μM ZnCl$_2$, was titrated with EGTA (Merck, USA) to obtain the following points: 50, 100, and 300 μM. The Zn stability constant of EGTA was taken equal to 0.63 nM[37]. At each point, the concentration of Zn-free TLR1-TIR was measured based on the intensity of W769 indole NH cross-peak. The concentration of Zn-free protein as a function of EGTA was approximated by the theoretical model, which is obtained by solving a system of equations, relating the stability constants and reagent/product concentration. The presence of two competing TLR1 zinc-binding sites that cannot be occupied simultaneously and are characterized by the Kd ratio of 1.0 was assumed.

**Protein crystallization, X-ray data collection, and structure solution**. TLR1-TIR was concentrated to 30 mg/ml in 20% v/v glycerol. ZnCl$_2$ was added at molar protein : Zn ratio of 1 : 0, 1 : 1, 1 : 10, and 1 : 20. Crystallization was done by a vapor diffusion method with HR2-122 screen (Hampton Research) at NT8 robot (Formulatrix), the protein : precipitant ratio was 1 : 1, and final drop volume was 300 nL. Plates were stored either at +4 °C or at room temperature (RT). Crystals appeared within 2–3 months at RT or 3–5 months at +4 °C, had a shape of square plates with typical size 200 × 200 × 15 μm$^3$ (Supplementary Fig. S5). Crystals were collected directly from crystallization drops, mounted on MiTeGen loops, and flash-frozen in liquid nitrogen. Single-crystal X-ray diffraction data were collected at 100 K ID23-1 ESRF λ = 0.972 Å. The data collection strategy was optimized in BEST[64].

X-ray datasets were collected for crystals in all used molar protein : Zn ratios of 1 : 0, 1 : 1, 1 : 10, and 1 : 20, and for both P6$_4$22 and P6$_2$22 space groups. Datasets in each condition have resolution in the range of 1.9–3.6 Å and all give similar protein structures. Deposited structures were solved from crystals obtained in 170 mM ammonium acetate, 85 mM sodium citrate tribasic dihydrate pH 5.6, 25.5% w/v polyethylene glycol 4000, 15% v/v glycerol (#9 from HR2-122 Hampton) at +4 °C for 1 : 0 and 1 : 1 molar protein : Zn ratios.

The data for Zn 1 : 1 condition were processed in the XDS software package[65]. The data for 1:0 Zn condition were processed with autoPROC pipeline[66]. The phase problem was solved by molecular replacement in Phaser[67] from PHENIX[68], where PDB ID 1FYV was used as a search model. The model was subsequently rebuilt in PHENIX.AutoBuild[69]; PHENIX.Refine and Coot[70] were used for model refinement. The quality of the resulting model was analyzed by PHENIX.MolProbity[71] and Quality Control Check web server (https://smb.slac.stanford.edu/jcsg/QC/). The crystallographic data collection and structure refinement statistics are given in Table 2. Structure of TLR1-TIR w/o Zn$^{2+}$ ions and with Zn$^{2+}$ ions have 93.04% and 98.09% amino acids in Ramachandran-favored region, and 6.96% and 1.91% amino acids in Ramachandran-allowed region, respectively.

**Computer modeling**. NMR structures were additionally relaxed with Rosetta Relax protocol[72]. Sampling of the L668-I679 region containing the BB-loop was performed using the Rosetta Loopmodel package with next-generation KIC protocol[73]. For every relaxed NMR structure, 100 independent configurations were generated, totaling in 2000. Analysis of contact frequencies between C667, C686, and other possible coordinators was performed with Prody[74] and Matplotlib[75].

Molecular modeling was carried out in GROMACS 2021.2 package[76]. Protein structures were capped with acetyl and N-methyl amide, and were placed in the center of the simulation box with the 1.5 nm offset between the molecule and box edges. Protein part was simulated with the amber14sb force field[77] with CUFIX corrections for electrostatic interactions[78] and additional parameters for Zn$^{2+}$ and Zn-coordinating residues[38]. Simulation box was filled with the explicit solvent molecules of the TIP3P water and salt concentration was adjusted to 0.15 M of NaCl and neutral total charge. Energy minimization, equilibration, and production procedures for protein in water are described elsewhere[79]. Production runs were performed with the following settings: temperature 300 K, timestep 2 fs, and trajectory length 500 ns. Every setup was independently repeated three times from

scratch. Simulations of systems with the backbone coordination mode were performed in two setups: unrestrained setup to test the viability of the coordination sphere and production restrained setup, to avoid rearrangements of the coordination sphere due on the larger timescales to the imperfections of the backbone coordination parameters. Harmonic restraining potential as implemented by the PLUMED[80] was applied to the distance I668_O - Zn at the value 2.3 Å with $\kappa = 150.0$.

**SEAP assay.** HEK Blue 293 cells, which were stably transfected with a SEAP reporter gene, were cultured in Dulbecco's odified Eagle's medium supplemented with 10% fetal bovine serum (FBS), penicillin (50 unit/mL), streptomycin (50 µg/mL), and 1× HEK blue selection. It should be noted that the SEAP reporter gene was placed under the control of NF-κB transcriptional response element. HEK Blue 293 cells were transfected with human TLR1 or TLR2 alone, and co-transfected WT or mutant human TLR1 and TLR2 using Lipofectamine 2000 (Invitrogen), according to manufacturer's instruction. After 48 h of transfection, cells were seeded at a concentration of $1 \times 10^5$ cells/mL. After 24 h incubation, medium was changed to Opti-MEM medium supplemented with 0.5% FBS, penicillin (50 unit/mL), streptomycin (50 µg/mL), 1% of non-essential amino acid (NEAA), and $Pam_3CSK_4$ (50 ng/ml) added to each well for 8 h. For Zinc-dependent assay, HEK Blue 293 cells co-expressing WT human TLR1 and TLR2 were seeded at the concentration of $1 \times 10^5$ cells/mL. After 24 h incubation, the medium was changed to Opti-MEM medium supplemented with 0.5% FBS, penicillin (50 unit/mL), streptomycin (50 µg/mL), 1% of NEAA, and $Pam_3CSK_4$ (50 ng/ml), and the indicated concentrations $ZnCl_2$ or TPEN for 8 h. NF-κB activity was detected by Phospha-Light™ SEAP Reporter Gene Assay System (Applied Biosystems, Foster, CA, USA) according to the manufacturer's instruction. All the experiments were repeated at least three times and analyzed using the directional Student's t-test.

**Quantitative reverse-transcriptase PCR.** HEK Blue 293 cells expressing human TLR1 or TLR2 alone and co-expressing WT or mutant human TLR1 and TLR2 were seeded at a density of $2 \times 10^5$ cells/mL in six-well plates. After 24 h treatment, total RNA was extracted by RNeasy Mini Kit. cDNA was synthesized by RT² Easy First Strand cDNA Synthesis Kit. Quantitative PCR (qPCR) was performed on a TOptical Real-Time qPCR Thermal Cycler (Analytik Jena, Thuringia, Germany) using the SYBR Green method. The data were analyzed by the ΔΔCt method. Sequences of primers used were as follows: β-actin (forward: 5′-TCGTGCGTGAC ATTAAGGAG-3′, reverse: 5′-ATGCCAGGGTACATGGTGGT-3′), TLR1 (forward: 5′-GCTGATCGTCACCATCGTTG-3′, reverse: 5′-GTCCACTGGCACACC ATCCT-3′), and TLR2 (forward: 5′-CCTCTCGGTGTCGGAATGTC-3′, reverse: 5′-GGCCCACATCATTTTCATATACC-3′).

**Statistics and reproducibility.** All experiments were performed three to four times independently and data are given as the mean ± SEM. The Student's t-test (directional) was used to calculate the statistical significance of differences between groups. Statistical analysis was carried out using the GraphPad Prism software 8.0 (GraphPad Software, La Jolla, CA, USA). The $p < 0.05$ was considered statistically significant.

**Reporting summary.** Further information on research design is available in the Nature Research Reporting Summary linked to this article.

## Data availability
NMR structure of the TLR1-TIR, peak lists, and chemical shifts were deposited to the PDB under the access code 7NT7 and to BMRB under the access number 34610. The structures for the TLR1-TIR crystallized with and without Zn were deposited in Protein Data Bank with accession numbers 7NUX and 7NUW, respectively. All other data are available from the corresponding author on reasonable request.

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

## Acknowledgements

This study was supported by the grants of Russian Foundation for Basic research (#20-34-70024, NMR analysis), National Natural Science Foundation of China (21877106 and 21807098), Pioneer Hundred Talents Program (CAS), and by the Ministry of Science and Higher Education of the Russian Federation (agreement #075-00337-20-03, project FSMG-2020-0003, and X-ray crystallography). We acknowledge the Structural Biology Group of the European Synchrotron for granting access to the synchrotron beamlines. We are grateful to E. Marin for the help with X-ray data collection, and A. Mishin and V. Gordeliy for the help with arranging crystallization and X-ray data collection. Molecular modeling was carried out using the equipment of the shared research facilities of HPC computing resources at Lomonosov Moscow State University.

## Author contributions

K.S.M., V.I.B, X.W., and S.A.G. designed the experiments. M.V.G. developed the protocol of protein production. V.A.L., M.V.G., and I.A.T. synthesized the recombinant proteins. V.A.L. performed the NMR experiments, single-point mutagenesis for NMR, and analyzed the data. I.A.T. performed the DLS experiments and analyzed the data. C.L. performed the NF-κB activity tests and mutagenesis, and analyzed the data. K.S.M and A.S.A. supervised the project. A.O.Z. performed computer modeling. A.P.L. did crystallization. M.B.S. prepared crystals and collected data. D.D.V. solved X-ray structures. V.I.B. supervised the crystallographic part of the work. K.S.M. wrote the paper with assistance from all the authors.

## Competing interests

The authors declare no competing interests.
