## [Transparent Peer Review File · Communications Biology]

Reviewers' comments:

Reviewer #1 (Remarks to the Author):

In their manuscript „Modulation of Toll-Like Receptor 1 Intracellular Domain Structure and Activity by Zn²⁺ Ions.” the authors present interesting data that can contribute to unraveling the complex interplay between zinc ions and TLR-signaling. Some improvements are recommended:

Statistical analysis in Fig 4: The authors used Student's t test for analyzing statistically significant differences in their data. However, t test can only be used for comparing experiments with two means; for three or more means a correction for multiple comparisons is required to avoid potential type 1 error. Statistical analysis has to be repeated with correct statistical methodology.

Typical zinc binding sites comprise 3-4 amino acids. Therefore, it is rather unusual that there seems to be only one relevant amino acid, C667, and this should not result in a binding site of nanomolar affinity. Therefore, the authors might discuss that there could be other, yet unidentified ligands (are there any carboxylated AAs or His in the vicinity that might be forming a potential binding site?), or could use bioinformatics to search for potential zinc-binding motifs. If all this should give no result, it should at least be discussed that zinc might also serve as an intermolecular bridging ion, with the remainder of the ligands being provided by the interacting protein.

Language editing: I am not sure if the authors want to emphasize the importance of TLRs the way they do (“Toll-like receptors (TLRs) play the essential role”; “Toll-like receptors (TLRs) are the important components...”), as these are certainly important receptors in innate immunity, but there are also other PRRs and many further proteins of comparable importance. So this is either a bit exaggerated, or of an unfortunately strong wording.

“Second, none of the studied so far TLR TIR domains...” is an awkward phrase and should be re-written.

Please make sure that all abbreviations are spelled out at first use (such as, MyD88, TIRAP, TRAF, ...).

Reviewer #2 (Remarks to the Author):

The authors describe the observation of Zn binding by the TLR1 intracellular TIR domain. The authors also present the solution structure of the TLR1-TIR domain including measurements of protein motion.

The work is presented well, although some language editing will be required. I also noted loss of characters in several places which will require attention prior to publication.

The central claim of the paper is supported well by the in vitro data. I do however have several queries that should be resolved prior to acceptance of this manuscript.

The authors do not mention the effect of calcium ions, which are highly abundant in the cell. An effect of calcium in competition with Zn at physiological concentrations must be resolved prior to further consideration of the significance of the claims of the manuscript. Similarly, the authors find that Co and Cu oxidise the Zn binding cysteine residues, but the mechanism for this is not discussed. It is also unclear what would happen under physiological concentrations of Zn and Cu. If physiological Cu concentrations oxidise the C667/C686 and thus inhibit Zn binding, it becomes difficult to see how the Zn binding is physiologically relevant.

I'm unfortunately not familiar with the functional assays performed but found it peculiar that the control NF-κB activity is different in the different panels with Pam3CSK4+ cells (panel B and C in particular). Considering that the ZnCl₂ or TPEN effects do not show a clear concentration dependence I'm hesitant to see this as compelling activity data. The mutagenesis data is more compelling but of course the mutated cysteine residues may be involved in any number of redox

reactions beyond the observed Zn binding.

For the NMR data I am concerned that the model free analysis is performed using data at a single field. It also appears odd that the order parameter is very low across the many helices of the protein. Given the propensity of these proteins to aggregate I would not trust any absolute measurements of motions, unless this was done at 2 dilute concentrations and at 2 fields. Personally, I don't see how this data (model free analysis) adds anything to the presented findings, beyond providing some support of the DLS data so perhaps presenting the measured data without the model free analysis would be more appropriate.

Finally, I note that in other TIR domains it has been found that higher order assemblies play an important role in downstream signalling, it would be interesting to know if Zn²⁺ leads to formation of TIR-1 filaments. I appreciate that in this case there is only a functional dimer present but this would provide insights into whether this TIR domain may initiate such a signalling complex.

RESPONSE TO THE REVIEWERS

First of all, we are very grateful to the reviewers for their expertise and opinion. We received several very constructive comments that, as we hope, will improve our manuscript. Below we provide a point-by-point response to all of the comments. All changes in the revised manuscript are highlighted in yellow.

Reviewer #1.

1. *"In their manuscript „Modulation of Toll-Like Receptor 1 Intracellular Domain Structure and Activity by Zn²⁺ Ions.” the authors present interesting data that can contribute to unraveling the complex interplay between zinc ions and TLR-signaling. Some improvements are recommended:*

- We thank the reviewer for his/her important comments. Below, we answer the queries of the reviewer.

2. *Statistical analysis in Fig 4: The authors used Student's t test for analyzing statistically significant differences in their data. However, t test can only be used for comparing experiments with two means; for three or more means a correction for multiple comparisons is required to avoid potential type 1 error. Statistical analysis has to be repeated with correct statistical methodology."*

- According to the statistical guides, *"Corrections for multiple comparisons may not be needed if you make only a few planned comparisons. The term planned comparison is used when: You focus in on a few scientifically sensible comparisons rather than every possible comparison. The choice of which comparisons to make was part of the experimental design. You did not succumb to the temptation to do more comparisons after looking at the data."* (https://www.graphpad.com/guides/prism/latest/statistics/stat_when_to_not_correct_or_2.htm). In our opinion, this is exactly the case of our experiments. We do not perform the scanning mutagenesis, we initially selected only 7 mutations, the selection was based on the protein structure, NMR data and amino acid properties. Thus, the multiple comparison correction is not needed for our experimental setup. To allow the readers and the reviewers to estimate the data, we redrew the Figure 4, which is now Figure 5 in the revised version (lines 273-280). In the new version, we use the dot-point format and indicate the exact p-value. In addition, we provide all the raw data and mRNA levels data (Figure S16) in supplementary materials.

Figure R1 (Figure 5 of the revised manuscript). A,B - NF-κB activity measured upon stimulation of HEK Blue 293 cells transfected with TLR1 and TLR2 genes with Pam₃CSK₄ with the addition of either 0-100 μM ZnCl₂ or 0-20 μM of TPEN to the culture (n=4). C - NF-κB activity measured for TLR1 mutants upon stimulation with Pam₃CSK₄ (n=3). Statistical significance is indicated as follows: *, **, *** and **** - p<0.05, p<0.01, p<0.001 and p<0.0001 with respect to the positive control, # and ##### - p<0.05 and p<0.0001 with respect to the negative control experiments. ns - denotes that changes with respect to the positive control are not significant.

3. "Typical zinc binding sites comprise 3-4 amino acids. Therefore, it is rather unusual that there seems to be only one relevant amino acid, C667, and this should not result in a binding site of nanomolar affinity. Therefore, the authors might discuss that there could be other, yet unidentified ligands (are there any carboxylated AAs or His in the vicinity that might be forming a potential binding site?), or could use bioinformatics to search for potential zinc-binding motifs. If all this should give no result, it should at least be discussed that zinc might also serve as an intermolecular bridging ion, with the remainder of the ligands being provided by the interacting protein."

- To answer, we turned to bioinformatics and computer modeling, the results are described in the new section of the revised manuscript: "**Computer modeling reveals two possible Zn-binding modes in TLR1-TIR BB-loop**" (lines 228-251). In brief, we generated an ensemble of possible BB loop conformations and identified H669, C667 and C686 as possible coordinators of Zn. Next we performed several MD simulations, starting from the various NMR structures of the generated ensemble. These simulations provided us two distinct Zn binding modes. First mode is through a classical CCH motif, and, most likely, is not relevant for the signaling. Second mode is peculiar, and employs C667 sidechain, backbone carbonyls and the oxygen of a water

molecule, which is additionally stabilized by the H-bonds with the backbone carbonyls of other BB loop residues. These two modes were found to stabilize two different conformations of TLR1 BB-loop, the second mode corresponds to the state, similar to the BB-loop of dimeric TLR10 TIR domain. We added a paragraph at the end of the discussion of the revised manuscript (lines 368-388). There, we analyze the results of computer modeling, NMR data and functional assay and express a hypothesis that Zn binding by TLR1-TIR stabilizes the conformation of the domain, which is capable of intermolecular interactions with TLR2 TIR domains or TIR domains of intracellular adaptor proteins (MyD88/TIRAP and etc). According to computer modeling, in all the complexes, the Zn ion is buried rather deep and is not accessible for intermolecular contacts, therefore, the role of zinc as an intermolecular bridging ion seems unlikely. According to this comment, we modified the Figures of our manuscript. Fig. 4 was split into two, and the new Fig 4. now includes the NMR data for the cysteine mutants and results of the *in silico* experiment, while Fig. 5 includes the results of the functional assays. Fig. 5 was additionally modified to comply with the policy of the journal. We would like to note that works on computer modeling were initiated earlier as a separate study and were not performed in a short term. We also added Arthur Zalevsky, who performed this work, to the list of authors.

Figure R2 (Figure 4 of the revised manuscript). Localization of TLR1-TIR Zn-binding site and the model of the Zn-bound state. **A** - an overlay of ^1H , ^{15}N -HSQC spectra (regions with the signal of W769 indole NH) of TLR1-TIR and its mutants C707A, C686A and C667A. Spectra were recorded before (in blue) and after (in red) the addition of $50\ \mu\text{M}$ ZnCl_2 to the

100 μ M sample of TLR1-TIR at 30 °C and pH 7.4. Signals corresponding to the Zn-free and two Zn-bound states of TLR1-TIR are assigned as apo, Zn1, and Zn2, respectively. **B** - snapshot from the simulation of the first coordination mode formed by C667-H669-C686. **C** - Closeup view of the coordination sphere for the C667-H669-C686 coordination mode. **D** - snapshot from the simulation of second coordination mode formed by C667 and I668_O. Additional coordinators might be represented by the R671_O and water molecules also coordinated by the backbone oxygen of the BB-loop residues. **E** - Closeup view of the Zn2 coordination sphere. **F** - comparison of the "extended" and "folded" BB loop conformations with the "native" BB-loop conformation in the TLR10 homodimer.

4. *"Language editing: I am not sure of the authors want to emphasize the importance of TLRs the way they do ("Toll-like receptors (TLRs) play the essential role"; "Toll-like receptors (TLRs) are the important components..."), as these are certainly important receptors in innate immunity, but there are also other PRRs and many further proteins of comparable importance. So this is either a bit exaggerated, of an unfortunately strong wording."*

- We modified the text to take into account this comment of the reviewer. Now in the abstract we state that "Toll-like receptors (TLRs) play an important role in the innate immune response" (line 22) and in the introduction we state that "Toll-like receptors (TLRs) take part in the innate immune response and may serve as targets ..." (line 35).

5. *"Second, none of the studied so far TLR TIR domains..." is an awkward phrase and should be re-written."*

- We rephrased this sentence. Now we state: "Second, all the studied TLR TIR domains do not homodimerize *in vitro*" (lines 60-61).

6. *"Please make sure that all abbreviations are spelled out at first use (such as, MyD88, TIRAP, TRAF, ...)."*

- We now spell out all the abbreviations, including MyD88, TIRAP, TRAF, Cryo-EM, RMSD, TCEP, EDTA, EGTA, TPEN, MOPS, SEAP.

Reviewer #2.

1. *"The authors describe the observation of Zn binding by the TLR1 intracellular TIR domain. The authors also present the solution structure of the TLR1-TIR domain including measurements of protein motion.*

The work is present well, although some language editing will be required. I also noted loss of characters in several places which will require attention prior to publication.

*The central claim of the paper is supported will by the *in vitro* data. I do however have several queries that should be resolved prior to acceptance of this manuscript."*

- We thank the reviewer for his/her positive assessment of our work. Below, we answer the queries of the reviewer.

2. *"The authors do not mention the effect of calcium ions, which are highly abundant in the cell. An effect of calcium in competition with Zn at physiological concentrations must be resolved prior to further consideration of the significance of the claims of the manuscript."*

- We thank the reviewer for raising the point, it was a serious flaw in our study. To answer this comment, we titrated the TLR1-TIR by CaCl₂ in the absence of Zn, similarly to other metals. It appears that calcium addition does not change the NMR spectra of

TLR1-TIR at all, suggesting that Ca does not interact with TLR1-TIR and the competition of zinc with calcium does not take place. We modified the main text, and we now mention that TLR1-TIR does not bind Ca ions (lines 183-184).

Figure R3. Overlay of ^1H , ^{15}N -HSQC spectra 100 μM TLR1-TIR recorded at 30 $^\circ\text{C}$, pH 7.4 and after addition 50 μM $\text{CaCl}_2/\text{ZnCl}_2$

3. "Similarly, the authors find that Co and Cu oxidise the Zn binding cysteine residues, but the mechanism for this is not discussed. It is also unclear what would happen under physiological concentrations of Zn and Cu. If physiological Cu concentrations oxidise the C667/C686 and thus inhibit Zn binding, it becomes difficult to see how the Zn binding is physiologically relevant."

- Indeed, we found that Co and Cu oxidize the C667-C686 disulfide bridge in the TLR1 TIR domain. It is known that Cu(II) can catalyze the disulfide oxidation, via the redox reaction, accompanied by the copper reduction to Cu(I). On the other hand, the similar reaction for cobalt, to our knowledge, has never been reported. Therefore, it is most likely that both Co and Cu bind to the TLR1, forcing it to adopt the conformation that favors the disulfide formation. To understand what happens under the physiological concentrations, we analyzed the kinetics of disulfide oxidation. At a concentration of both TLR1 and Co/Cu of 100 μM , the reaction takes 1-2 hours, and it is slower in case of Cu than in the case of Co. NMR spectra of TLR1-TIR reveal only two states - the unbound state and disulfide-oxidized state of TLR1. The latter observation implies that the binding constant of Co and Cu is above 100 μM , otherwise we would observe the metal-bound state and disulfide-crosslinked state. The first observation suggests that at the native concentrations of Co and Cu, which are at least several orders of magnitudes lower than 100 μM , the disulfide oxidation would take proportionally longer times. Therefore, under the physiological concentrations of metals, the Co/Cu bound states of TLR1 would be low-abundant and the disulfide oxidation would run very slowly. This is additionally supported by the results of mutagenesis. One would expect C667A and C686A mutations to cause the similar effect on TLR1 activity if the disulfide is formed under physiological conditions, while we observe the opposite - one mutation has an effect, while the other does not. To take into account the possibility of Co/Cu-

induced disulfide oxidation, we modified the discussion section of our manuscript (lines 296-311).

4. *"I'm unfortunately not familiar with the functional assays performed but found it peculiar that the control NF-κB activity is different in the different panels with Pam3CSK4+ cells (panel B and C in particular)."*

- As biological assays cannot be consistent every time, the validity of the results can be demonstrated once three or more repeated experimental trials are proven to be statistically significant. Thus, while Panel B and panel C of Figure 4 (now panels A and B in Figure 5) were not performed simultaneously the difference in NF-κB activities in the different panels with Pam3CSK4 is normal.

5. *"Considering that the ZnCl₂ or TPEN effects do not show a clear concentration dependence I'm hesitant to see this as compelling activity data. The mutagenesis data is more compelling but of course the mutated cysteine residues may be involved in any number of redox reactions beyond the observed Zn binding."*

- To improve the reliability of our results, we re-examined the ZnCl₂ and TPEN effects for the activity of NF-κB. We made the additional repeats and included more concentration points. The new data may be seen in Figure 5 of the revised manuscript, and the concentration dependence is now clearly observed (lines 274-280). We modified the Results section to include the new data (lines 260-262). The revised figure may be found in the response to reviewer #1.

We understand the concerns of the reviewer, however, it is impossible to find a functional assay that would provide completely undoubtful evidence in favor of our hypothesis. In our opinion, we provide as much data as possible. Effects of ZnCl₂ or TPEN reveal that Zn is somehow involved in the TLR1 activation, while the mutagenesis data correlate well with the results of NMR studies. We analyze several possible explanations of the effect of C667A mutations, and find that the Zn-binding hypothesis is the most probable. In the revised version we use computer modeling to find the possible structure of two Zn-bound states of TLR1-TIR and provide the structure-based explanation of how Zn binding may support the TLR1 activation. We hope that the reviewer will find the evidence in the revised version more compelling. We agree that we omitted the possibility that C667 may be involved in redox reactions. However, the TIR domain is intracellular and the cytoplasmic environment is highly reducing. Furthermore, redox reactions were never reported to participate in TLR activation. Thus, we consider this explanation as highly unlikely. We modified the discussion of the manuscript to take into account the redox reaction as a possible explanation of the observed effects of C667A mutation (lines 359-362).

6. *"For the NMR data I am concerned that the model free analysis is performed using data at a single field. It also appears odd that the order parameter is very low across the many helices of the protein. Given the propensity of these proteins to aggregate I would not trust any absolute measurements of motions, unless this was done at 2 dilute concentrations and at 2 fields. Personally, I don't see how this data (model free analysis) adds anything to the presented findings, beyond providing some support of the DLS data so perhaps presenting the measured data without the model free analysis would be more appropriate."*

- We provide the results of model-free analysis in order to additionally validate our solution NMR structure. It contains several regions of low convergence, and relaxation

data confirms that these regions are actually mobile and this is not the problem of NMR data quality. In this regard, we use the results of model-free analysis as qualitative indicators of internal mobility and do not aim to elucidate the exact characteristic times of internal motions. For this purpose, the analysis, performed at a single field, is in our opinion sufficient, even if the low-populated dimeric state is present. The raw NMR relaxation data are not that illustrative. Nonetheless, we had initially recorded the NMR relaxation data at two concentrations: 500 and 150 μM , and only the 500 μM data were placed in the manuscript. We additionally processed the data at 150 μM and obtained similar results. We now provide these data in Figure S3 of the revised version (see Figure R4 below).

Figure R4 (Figure S3 of the revised manuscript). NMR relaxation parameters of ^{15}N nuclei (rates of longitudinal (R_1) and transverse (R_2) relaxation, heteronuclear equilibrium NOE ($^1\text{H}, ^{15}\text{N}$ NOE)), and results of the model-free analysis ($S_0^2 S_1^2$, R_{ex}) are provided for the aminoacid residues of TLR1-TIR at two concentrations (blue corresponds to 3.3 mg/ml and orange - to 10.3 mg/ml).

7. *"Finally, I note that in other TIR domains it has been found that higher order assemblies play an important role in downstream signalling, it would be interesting to know if Zn²⁺ leads to formation of TIR-1 filaments. I appreciate that in this case there is only a functional dimer present but this would provide insights into whether this TIR domain may initiate such a signalling complex."*

- Indeed, TIR domains are capable of forming assemblies and this behaviour has been well defined for MAL-TIR, which spontaneously and reversibly forms filaments in vitro (10.1038/nsmb.3444, 10.1038/s41467-021-22590-6). It also forms co-filaments with TLR4-TIR domains and induces the assembly of MyD88-TIR complex. In addition, the formation of the myddosome (MyD88-IRAK4-IRAK2) in vitro was described without the participation of TLR-TIR (10.1038/nature09121, 10.1016/j.str.2020.01.003). Thus, most likely, pre-filaments can exist in cells by default in the absence of TLR1/Zn complex and the activation of TLR-TIR leads to signalosome assembly. Based on our computer modeling data (new section on the revised manuscript) the Zn ion is not accessible for intermolecular contacts and we assume that Zn binding stabilizes the TLR1-TIR conformation, which is required for interactions with adapter proteins. Confirmation of this hypothesis will require a separate study with direct detection, for example by cryo-EM, because our NMR data provide information only about the monomer of TLR1-TIR.

REVIEWERS' COMMENTS:

Reviewer #1 (Remarks to the Author):

In the revised version of their manuscript „Modulation of Toll-Like Receptor 1 Intracellular Domain Structure and Activity by Zn²⁺ Ions.“ the authors underwent (in most cases) great efforts to address the comments raised in my previous review.

With regard to statistical analysis, I do not completely agree with the authors' evaluation of their analyses as "a few planned comparisons". E.g., in Fig 5A/B, ALL zinc/TPEN concentrations are being compared to Pam3CSK4. Of course, one could argue that all of these were planned, otherwise the data points would not have been investigated, but it is pretty much a general testing of all (not a selected few) means against Pam3CSK4. I do not think that this is what the experts at graphpad had in mind when they were writing this recommendation. For this case a oneway ANOVA with Dunnett's Posthoc test would probably have give very similar significances and have been more appropriate. Also, the targets in 5C were certainly selected in advance. However, there are just a few approaches where this is not the case (mostly non-targeted ***omics) and correction for multiple comparisons is certainly not just recommended for those. Most experiments are typically planned with some rationale in mind with targets that are carefully selected. Yet, the current way to present the data is quite clear, so if this is how the authors chose to proceed, it does not make the paper unacceptable.

In general, the manuscript has certainly been improved a lot and I can recommend publication on Communications Biology in its present form.

Reviewer #2 (Remarks to the Author):

The authors have addressed my concerns sufficiently.

RESPONSE TO THE REVIEWERS

We are very grateful to the reviewers for their expertise, which helped us to improve our manuscript.

Reviewer #1.

In the revised version of their manuscript „Modulation of Toll-Like Receptor 1 Intracellular Domain Structure and Activity by Zn²⁺ Ions.” the authors underwent (in most cases) great efforts to address the comments raised in my previous review.

*With regard to statistical analysis, I do not completely agree with the authors' evaluation of their analyses as “a few planned comparisons”. E.g., in Fig 5A/B, ALL zinc/TPEN concentrations are being compared to Pam3CSK4. Of course, one could argue that all of these were planned, otherwise the data points would not have been investigated, but it is pretty much a general testing of all (not a selected few) means against Pam3CSK4. I do not think that this is what the experts at graphpad had in mind when they were writing this recommendation. For this case a oneway ANOVA with Dunnett's Posthoc test would probably have give very similar significances and have been more appropriate. Also, the targets in 5C were certainly selected in advance. However, there are just a few approaches where this is not the case (mostly non-targeted ***omics) and correction for multiple comparisons is certainly not just recommended for those. Most experiments are typically planned with some rationale in mind with targets that are carefully selected. Yet, the current way to present the data is quite clear, so if this is how the authors chose to proceed, it does not make the paper unacceptable.*

In general, the manuscript has certainly been improved a lot and I can recommend publication on Communications Biology in its present form.

- We thank the reviewer for his/her positive assessment and important comment, and we will keep in mind these approaches in the future.

Reviewer #2.

The authors have addressed my concerns sufficiently.

- We thank the reviewer for his/her positive assessment of our work.